# Dietary Interventions in Inflammatory Bowel Disease

**DOI:** 10.3390/nu14204261

**Published:** 2022-10-12

**Authors:** Małgorzata Godala, Ewelina Gaszyńska, Hubert Zatorski, Ewa Małecka-Wojciesko

**Affiliations:** 1Department of Nutrition and Epidemiology, Medical University of Lodz, 90-752 Lodz, Poland; 2Department of Digestive Tract Diseases, Medical University of Lodz, 90-153 Lodz, Poland

**Keywords:** inflammatory bowel disease, dietary interventions, nutrition, intestinal microbiota, lactose-free diet, low FODMAP diet, gluten-free diet, specific carbohydrates diet, anti-inflammatory diet, Mediterranean diet

## Abstract

Inflammatory bowel disease, which primarily includes ulcerative colitis and Crohn’s disease, is a group of chronic diseases of the gastrointestinal tract. Mainly affecting young people, it is characterized by periods of exacerbation and remission. In recent years, there has been an increase in the prevalence of inflammatory bowel disease worldwide, including Poland. The potential impact of nutrition and selected dietary components that are directly or indirectly involved in the pathogenesis of intestinal lesions in IBD is not fully clear. Evaluating the impact of diet on the course of IBD is very complex due to the fact that regardless of a dietary model adopted, each one is based on consumption of many different food groups which affect one another. However, the growing need to produce dietary recommendations for these patients has prompted the International Organization for the Study of Inflammatory Bowel Disease (IOIBD) to develop nutrition guidelines for the patients. The present paper characterizes the dietary models most commonly discussed in research studies and their potential impact on IBD activity.

## 1. Introduction

Inflammatory bowel disease (IBD), which primarily includes ulcerative colitis (UC) and Crohn’s disease (CD), is a group of chronic diseases of the gastrointestinal tract. Mainly affecting young people, it is characterized by periods of exacerbation and remission [1,2]. These conditions, whose incidence in the population has increased in recent decades, are among the idiopathic, chronic and recurrent inflammations of the gastrointestinal tract [2]. Confirmed mechanisms underlying the development of these diseases include the dysregulation of the immune system and changes in the intestinal microflora [1,2]. Additionally, factors that contribute to their development are oxidative stress and a defect in the intestinal mucosal barrier [1,3]. In recent years, there has been an increase in the prevalence of inflammatory bowel disease worldwide, including Poland. It is estimated that approximately three million people in Europe suffer from the disease, however, these data may be incomplete [1]. The cause of the observed epidemiological trends in inflammatory bowel disease is unknown. Certainly, the progressive degradation of the environment, poor quality of food and overuse of certain drugs, especially antibiotics, contribute to these phenomena. The significant role of environmental factors is evidenced by the example of Asian countries, where these diseases used to be rare. Nevertheless, with the westernization of life, their incidence is now growing rapidly [1,3,4].

The potential impact of nutrition and selected dietary components that are directly or indirectly involved in the pathogenesis of intestinal lesions in IBD is not fully clear. Available research results are inconclusive, often burdened by methodological limitations and based on small groups [3,4,5,6,7,8]. However, the growing need to produce dietary recommendations for these patients has prompted the International Organization for the Study of Inflammatory Bowel Disease (IOIBD) to develop nutrition guidelines for the patients [5].

Evaluating the impact of diet on the course of IBD is very complex due to the fact that regardless of a dietary model adopted, each one is based on the consumption of many different food groups which affect one another. There are reports on the elimination of certain foods and nutrients that may exacerbate symptoms or maintain remission of the disease [6,7,8]. It is suggested that individualized recommendations should be developed, taking into account the patient’s nutritional status, their own experience and the course of the disease. An unquestionable argument for the beneficial effects of dietary interventions is the fact that they improve the patient’s quality of life, lessen the severity of disease symptoms such as diarrhea and abdominal pain, and reduce the incidence of complications [9,10,11].

The present paper characterizes the dietary models most commonly discussed in research studies and their potential impact on IBD activity.

### Intestinal Microbiota, Nutrients and IBD

An important aspect of nutrition is the impact of selected dietary components on the intestinal microbiota, which plays an important role in the onset and course of IBD [1,3,4,5,6,7,12]. Studies indicate that it is mainly a high-fat diet, rich in processed foods, refined sugars and red meat (Western diet) that contributes to the negative modification of the microbiota [12,13,14,15]. A study by Di Paola et al. showed that individuals following a low-fat, high-fiber diet had a more diverse microbiota and a lower proportion of pathogens than those on the Western diet, due to the low proportion of complex carbohydrates and dietary fiber [16]. Moreover, it has been shown that the combination of high fat and sugar, which is characteristic of the Western diet, leads to intestinal dysbiosis in mice through an increase in the amount of *Bacteroides* spp. and *Ruminococcus torques*. What is also observed, is an increased amount of *Enterobacteria*, *Bilophila* spp., *Alistopes* spp. and *Akkermansia* spp., and a decreased amount of *Bifidobacterium*, *Lactobacillus*, *Prevotella*, *Roseburia*, as well as *Eubacterium rectale* and *Ruminococcus bromii* [17]. The Western diet may lead to metabolic endotoxemia by elevating the amount of endotoxin-producing bacteria and increasing intestinal permeability [13,14,15]. This phenomenon results from the high fat content of the diet, though it may also be influenced by an insufficient intake of insoluble fiber, which is attributed to playing a key role in the formation of normal intestinal microbiota [18,19]. Dietary fiber increases the diversity of the intestinal microbiota. This is related to the fact that polysaccharides that are not broken down by human digestive enzymes are fermented by intestinal bacteria, representing the genera of *Bacteroides*, *Roseburia*, *Bifidobacterium*, *Faecalibacterium* and *Enterobacteria*. The product of anaerobic fermentation is short chain fatty acids (SCFAs): acetate, propionate and butyrate in a ratio of 60:25:15. SCFAs are mainly produced by the bacteria of the *Bifidobacterium* and *Lactobacillus* genera. These acids play an integral role in maintaining immune homeostasis, functioning as signaling molecules that link the immune, nervous and gastrointestinal systems [20]. SCFAs also participate in maintaining the continuity of intestinal epithelial tissue, provide a rich source of energy, both for bacteria and colonocytes, and counteract the development of diseases, including guiding potentially cancerous cells into the apoptosis pathway [21]. The type of SCFAs formed mostly depends on the substrate supplied (the type of fiber) and the predominant type of intestinal bacteria, so the type of food consumed significantly affects health. Thus, fiber acts as a prebiotic, i.e., a substance not digested by humans, whose fermentation products stimulate the growth of probiotic bacteria in the intestines. Therefore, dietary models using plant-based ingredients (vegetarian, Mediterranean diets) contribute to the growth of commensal bacteria and levels of SCFAs which are the main source of energy for colonocytes [20,21]. Studies have shown that reducing the amount of carbohydrates in the diet leads to a decrease in the intestinal content of butyrate-producing bacteria (*Roseburia* spp., *Eubacterium rectale*, *Bifidobacterium* spp.), however, it does not stimulate changes in the amount of *Bacteroides* spp. These are unfavorable changes, since butyrate is the primary energy substrate and affects the resistance of epithelial cells against harmful agents, whereas, consuming a larger amount of carbohydrates results in an increase in the total number of bacterial cells [22,23,24]. Additionally, plant-based diets contain large amounts of polyphenols. Polyphenols exhibit effects similar to those of conventional drugs, modulating cellular signaling pathways by inhibiting the expression of pro-inflammatory cytokines and inflammatory mediators or transcription factors. Owing to these properties, they contribute to the reduction of inflammation in patients with IBD. Additionally, increased dietary polyphenols have been shown to help restore the integrity of the intestinal barrier by reducing lipopolysaccharide-induced intestinal permeability disorders [19]. Lipopolysaccharide is a building component of Gram-negative bacteria present in the gastrointestinal tract. In the intestinal lumen, inorganic microparticles from the diet, such as food additives, combine with lipopolysaccharides in the bacterial cell wall to form antigenically active substances which can modulate local and generalized immune responses. The consumption of foods rich in sulfur compounds (pyrosulfite, sulfur dioxide) and highly processed foods (with emulsifiers, synthetic colors and preservatives), contributes to the rapid growth of sulfate-reducing bacteria (SRB).

Some reports have identified similarities in the composition of the microbiota in UC and CD, while others point to differences depending on the diagnosis [25,26,27]. The involvement of the intestinal microbiota in the onset, development and maintenance of the inflammatory process in IBD no longer raises any doubts. It can be considered in two aspects, i.e., as a specific, invasive source of pathogens that cause inflammatory changes (theory of infection) or as a source of antigens that induce an abnormal, excessive immune response with the release of inflammatory mediators leading to tissue damage [28].

There is no evidence of a specific pathogen causing IBD, although many reports have emphasized the involvement of the *Mycobacterium avium subspecies paratuberculosis* (MAP) bacteria in the pathogenesis of the disease [29,30,31,32,33,34]. This bacillus causes granulomatous enteritis in ruminants, which histologically resembles CD in humans [31]. Although MAP was not proven to infect humans from animals, MAP-specific genetic material has been found more frequently in intestinal tissue collected from CD patients than in healthy humans, as well as in UC patients [32]. Other pathogens that are potential etiologic agents of IBD include the measles virus, microorganisms such as *Pseudomonas*, *Chlamydia*, *Yersinia pseudotuberculosis*, *Listeria monocytogenes*, or adjacent strains of *Escherichia coli* [33]. However, the finding of the presence or increased titers of antibodies against particular types of viruses or bacteria in the serum of IBD patients does not prove their role as a causative agent of the disease.

The saprophytic intestinal flora, which are important for the proper functioning of the host body, remain in a state of homeostasis with the intestinal immune system. Disturbances of protective immune and regulatory mechanisms can induce abnormal immune responses and the development of inflammatory reactions leading to damage to the intestinal tissue. In the course of IBD, very significant quantitative and qualitative differences are found in the composition of the intestinal microflora of IBD patients, as compared to healthy individuals [9,10]. The number of *Enterobacteriacae bacteria*, including *E. coli*, is significantly increased, while the proportion of *Clostridium* group IV and XIV is reduced. This is very important, since *Clostridium* bacteria provide SCFAs. Lower fecal SCFA levels in IBD patients, especially in those with UC, along with an impaired utilization of SCFA due to an increased production of hydrogen sulfide by an increased number of sulfate-reducing strains, probably contribute to the disease [8,9]. Studies have shown the presence of SRB in stool samples collected from IBD patients [35,36,37]. During the growth of these bacteria, sulfur is used to form mucosal bridges in the intestine. Mercaptoids formed during sulfite reduction, along with sulfides, affect fatty acid oxidation in colonocytes. An abnormal intestinal mucus is formed due to the production of highly toxic hydrogen sulfide, which damages the intestinal epithelium. Many authors indicate that by this mechanism, sulfate-reducing bacteria contribute to the development of IBD [35,37,38].

It is still impossible to fully determine in every case whether the differences in microbial composition and function found in this group of patients are the primary factors involved in the pathogenesis of IBD, or are secondary consequences of inflammatory reactions. Nevertheless, modifications achieved, for example, with the use of probiotics, prebiotics or certain antibiotics, are promising therapeutic prospects. There are available studies evaluating the effects of antimicrobial peptides (AMPs) on the intestinal microbiome [39,40,41]. Due to continuous exposure to bacterial agents and food antigens, the intestinal mucosa is equipped with a number of mechanisms to support its barrier functions. One of these is the production of AMPs which have the ability to shape the microbiome composition [42]. Moreover, AMPs can be applied in therapy as standalone antimicrobial drugs, offering an alternative to antibiotics. They can also be used in combination with antibiotics and synergistically enhance their antimicrobial effects. They also exhibit immunomodulatory effects by interacting with Toll receptors, thereby regulating the expression of cytokines or affecting the exposure of adhesion proteins [39,40,41,42]. An important element observed in the pathogenesis of IBD is the dysfunction of Toll-like receptors (TLRs). These structures are responsible for recognizing bacterial antigens which come into contact with the intestinal mucosa [40,41]. In the case of infection with pathogenic bacteria, they trigger a cascade of immune responses. They are also responsible for the homeostasis of the intestinal mucosa and proper functioning of the intestinal barrier. The involvement of endothelial adhesion molecules in the pathogenesis of IBD is also the subject of current research. An important role in the development of inflammation in these diseases has been attributed to vascular cell adhesion molecule 1 (VCAM1), very late antigen 4 (VLA4) and intracellular adhesion molecule 1 (ICAM1), among others. Their expression in the endothelium is stimulated by pro-inflammatory cytokines. These molecules, with the help of chemoattractants, attract leukocytes (mainly neutrophils and monocytes) from the peripheral blood to the site of inflammatory infiltrate formation [39,40,41]. Currently, endothelial adhesion molecules, due to their important involvement in the pathogenesis of inflammatory bowel disease, are considered as a target for therapy with biologic drugs [43].

## 2. Lactose-Free Diet

Lactose is a disaccharide made up of a glucose and galactose molecule, found in mammalian milk (cow, goat and sheep). The underlying cause of lactose intolerance, a condition in which clinical symptoms are causally related to lactose consumption, is a lack or deficiency of lactase, a brush border enzyme of the small intestine. It catalyzes the hydrolysis of lactose to simple sugars which are absorbed in the small intestine. Lactose is not absorbed, but it increases the osmolality of the intestinal contents, which causes diarrhea, and bacterial fermentation produces short-chain organic acids as well as hydrogen, carbon dioxide and methane, which results in bloating [44,45,46,47]. The most common cause of lactose intolerance symptoms is the physiological process of decreasing lactase activity with age, the so-called primary adult-type hypolactasia. Symptoms of lactose intolerance generally do not occur until lactase activity falls below 50%. Secondary hypolactasia can be triggered by a variety of factors, including viral and bacterial infections or the exacerbation of Crohn’s disease, with lactase activity rising again once these have subsided [48,49].

The lactose content of milk and milk products is similar, however, studies have confirmed that lacto-fermented products (such as yogurt) are better tolerated [50,51,52]. This is due to the endogenous activity lactase of the microorganisms in them. Yogurts contain bacteria of the genera *L. bulgaricus* and *S. thermophilus*, which are involved in the hydrolysis of lactose both during fermentation processes and after consumption. Therefore, it is believed that a yogurt containing *108 S. thermophilus* and *L. bulgaricus* bacteria/mL is as effective as enzyme supplements [53,54]. The fermentation process also occurs during the process of cheese maturation, which is why yellow cheeses contain no or trace amounts of milk sugar. Lactose-free products, on the other hand, are devoid of lactose through its enzymatic hydrolysis [51].

The dietary elimination of lactose is a form of nutritional management often indicated in studies among patients with IBD [52,53,54,55,56]. However, the rationale for its implementation is controversial, as it has not been established beyond doubt that IBD patients are more likely to be lactose intolerant than the general population. A study by Eadal et al. showed the prevalence of lactose intolerance in 70% of IBD patients, which was confirmed by a hydrogen test and genetic testing, as well as manifested by clinical symptoms [57]. In contrast, in a study by Jasielska et al., the prevalence of lactose intolerance in children with CD and UC was 23.2% and 22.6%, respectively, and did not differ significantly between sick and healthy individuals [44]. In another Polish study, the prevalence of lactose intolerance was estimated to be about 30% in the healthy population [58]. In a study by Adler et al., the prevalence of lactose intolerance in healthy and CD groups was 30% and 34%, respectively, and those differences were not statistically significant [59]. Similar data were obtained in a study by Büning et al. in which the prevalence of lactose intolerance was not significantly different between healthy subjects, patients with IBD and their relatives [60].

There are reports on a potential influence of milk and dairy products on inflammatory processes and the development of IBD [61]. There are papers from the 1990s confirming a correlation between high dairy consumption and a high incidence of IBD [62,63]. A study by Kitahora et al. analyzed the influence of genetic and environmental factors on the incidence of UC in the Japanese population. The study evaluated changes in the prevalence of UC occurring between 1966 and 1975, and their relationship to the dietary behavior of the Japanese population. It showed that with higher daily intake of dairy products, the incidence of UC increased as well [62,63]. In contrast, reports in recent years have failed to confirm the association between lactose intake and the incidence of IBD [64,65] or they even demonstrated the protective effect of dairy products by reducing the severity of symptoms in this group of patients [66,67,68]. In a study evaluating the intake of food groups in relation to UC activity, milk and cheese were among the products that alleviated the symptoms [66]. In a study by Octoratou et al., yogurts were products that reduced the risk of CD [67]. In contrast, in a study by Opstelten et al., a lower intake of milk, but a higher intake of cheese were observed in patients with IBD, as compared to healthy subjects [68]. According to the latest recommendations of the European Crohn’s and Colitis Organization (ECCO) and the European Society for Clinical Nutrition and Metabolism (ESPEN), introducing any elimination diet in IBD patients, without blood test-confirmed food intolerance, is unreasonable and harmful [69,70].

Due to the enormous nutritional value of milk and dairy products, they constitute the basis of rational nutrition and an integral part of a daily diet. Their elimination is associated with a high risk of calcium and vitamin D deficiency, which can lead to skeletal demineralization and, in extreme cases, to malnutrition, including protein malnutrition, which greatly hinders pharmacotherapy and worsens prognosis. This is particularly important in patients with IBD, in whom osteopenia and osteoporosis result from, among others, recurrent inflammation and the use of pharmacotherapy (steroids) [71,72]. As demonstrated in studies, a significant group of IBD patients believes that eliminating lactose from their diet will reduce symptoms and maintain IBD remission, despite the absence of blood test-confirmed lactose intolerance. In addition, it has been shown that a significant number of patients permanently cut out dairy products, regardless of periods of exacerbation and remission of the disease, for fear of the recurrence of symptoms [53,54]. Thus, when diagnosing and treating patients with IBD, it is crucial to confirm lactose intolerance, which prevents the unnecessary implementation of elimination diets. Molecular testing of the lactase gene polymorphism and the hydrogen/hydrogen–methane breath test are recommended. The latter, due to its high sensitivity and specificity, as well as its availability, is considered the gold standard in the diagnosis of lactose intolerance [73].

## 3. Gluten-Free Diet

Gluten is a protein commonly found in wheat, rye and barley kernels. Gluten consists of two types of proteins: gliadin and glutenin. Gliadin exhibits significant viscosity, while glutenin is elastic. Both fractions are found in similar proportions in the grain’s endosperm. Due to its high water-binding capacity and its corresponding elasticity and malleability, gluten forms viscoelastic membranes that maintain the proper spongy consistency of dough during fermentation and baking. Additionally, gluten, as a good carrier of flavors, is often used by manufacturers as a food additive [74,75].

A gluten-free diet is indicated for celiac disease, Duhring’s disease and wheat allergy. A gluten-free diet is not recommended in patients with IBD, however, studies have shown that it is used by patients to alleviate symptoms, despite the lack of objective indications [54,56,76]. Although both visceral disease and IBD promote dysregulation of the immune response, leading to permanent inflammation, changes in the intestinal microflora caused by the absence of gluten have been demonstrated. These manifest themselves in lower concentrations of *Bifidobacterium* spp. and *Lactobacillus* spp. and, consequently, lead to a decrease in short-chain fatty acids and their positive impact on the immune function of the body [3,74,77]. Studies have verified the effect of a gluten-free diet on the occurrence of gastrointestinal symptoms in patients with IBD. In the study by Herwarth et al., the majority of IBD patients following a gluten-free diet reported a less frequent occurrence of symptoms such as abdominal pain, diarrhea and bloating as compared to the patients on a regular diet. Moreover, when on the diet, respondents reported their partial discontinuation of pharmacotherapy due to clinical improvement [78]. Similar data were also obtained in other studies, in which IBD patients experienced a subjective alleviation of the disease symptoms when following a gluten-free diet, in the absence of diagnosed celiac disease, however, it was not accompanied by an objective improvement in terms of a reduction in CDAI or a reduction in the frequency of disease exacerbations [79,80]. Similarly, a study by Weaver et al. found no differences in disease activity or hospitalization rates between patients with IBD on a gluten-free and standard diet [3]. A recently published summary of three cohort studies, i.e., the Nurses’ Health Study (NHS), the NHS II and the Health Professionals Follow-Up Study (HPFS), showed no effect of a gluten-free diet on the risk of CD and UC [81].

There is still no scientific evidence that would be an indication for eliminating gluten from the diet of IBD patients. Often, the improvement in health condition when following a gluten-free diet is attributed to the simultaneous elimination of FODMAPs that are found in large quantities in gluten-containing products. This may contribute to the decreased severity of the accompanying functional symptoms. Although animal studies have shown that gluten is a factor involved in the adverse modulation of immune pathways in the small intestine, and other wheat proteins (amylase and trypsin inhibitors) can induce local inflammation, there is no evidence of a therapeutic effect of a gluten-free diet on IBD once gluten is eliminated [82,83].

An important aspect of introducing a gluten-free diet in patients with IBD is the awareness of its lifelong use. While in patients with IBD and concomitant visceral disease such a diet is mandatory, in patients with isolated IBD, it can negatively affect their general health condition, and cause mood swings and a tendency towards depression. The mechanisms for the development of psychiatric disorders in celiac disease remain unknown. Theories about them include disorders of the brain–gut axis or the emotional and social consequences of celiac disease diagnosis [3]. Thus, on the one hand, it has been suggested that a proper diagnosis and implementation of a gluten-free diet, and the absence of gastrointestinal symptoms protect against psychiatric disorders, while on the other hand, the need to adhere to the diet worsens patients’ quality of life, causes social isolation and may itself contribute to mood disorders [3,78].

## 4. Low FODMAPs Diet

The low Fermentable Oligosaccharides, Disaccharides, Monosaccharides And Polyols (FODMAPs) diet limits the intake of fermentable mono-, di- and oligosaccharides, and polyols. Products rich in FODMAPs pass into the intestine unchanged, and due to their osmotic properties, they contribute to increased water absorption in the small intestine. Additionally, in the large intestine, they are fermented by intestinal bacteria along with the production of excessive gas causing pain, discomfort and bloating [61,84].

Studies have shown that easily fermentable, poorly absorbed and high osmotic pressure carbohydrates, which include fructose, lactose, fructans and polyhydroxy alcohols (sorbitol, mannitol, maltitol, xylitol), can exacerbate symptoms in healthy individuals and patients with irritable bowel syndrome [75,85,86,87,88,89,90]. There are also papers indicating the validity of a low FODMAP diet in patients with UC and CD. A study by Anderson et al. pointed out that IBD patients eliminated FODMAP-rich foods on their own, especially during symptom exacerbation, and the decision to cut out such products was taken much more frequently than among healthy subjects and patients in remission [91]. In a six-week follow-up, Prince et al. showed improvement in most IBD-related symptoms such as abdominal pain, bloating and diarrhea in relation to a low FODMAP diet in 72% of patients with CD and 78% of patients with UC without active inflammation [92]. Additionally, Bodini et al., in a study conducted in a group of patients with IBD, found lower calprotectin values, reduced disease activity and improved quality of life in these patients as measured by the Short IBD Quality of Life Questionnaire (SIBDQ) after a FODMAP-restricted diet [93]. In contrast, a study by Melgaard et al. showed a reduction in abdominal pain and bloating after the introduction of a low FODMAP diet in UC patients, however, symptoms returned after both placebo and FODMAP provocation. The authors of the study explained the dynamics of the changes in these patients by the placebo and nocebo effects [94].

The low FODMAPs diet is also discussed in terms of treating small intestinal bacterial overgrowth (SIBO), which accompanies IBD patients. SIBO is a syndrome of gastrointestinal symptoms caused by the presence of an excessive amount of bacteria in the small intestine that cause dysbiosis [95]. Structural abnormalities such as small bowel diverticula, anatomical anomalies, conditions following intra-abdominal surgeries (post-operative fistulas, intraperitoneal adhesions, lesions of the ileocecal valve lesions) or CD predispose a patient to SIBO [96]. An abnormal intestinal microbiota is associated with the presence of fermentable products, so their elimination from the diet contributes to reducing bacterial growth and proliferation. Data on the dietary management of SIBO are similar to those recommended for irritable bowel syndrome (IBS) [75,95,97,98,99]. Dietary factors are now believed to play a key role in the development of IBS symptoms in up to 84% of patients [95]. A meta-analysis of papers on the use of low FODMAP and gluten-free diets in IBS did not provide sufficient evidence to recommend a gluten-free diet for patients with IBS. However, an effect of a low FODMAP diet on relieving the symptoms accompanying IBS was demonstrated [75]. The main reason for the rapid induction of symptoms after FODMAP intake and their alleviation within a few days of starting a low FODMAP diet is probably an altered mechanoreceptor stimulation caused by the dilatation of the lumen of the digestive tract [90]. Diets rich in soluble fiber increase the production of short-chain fatty acids while inhibiting potentially invasive bacteria such as Escherichia coli and other species of *Enterobacteriaceae* [97]. Soluble fiber supplementation is strongly recommended in the American and Polish guidelines for the treatment of IBS, including in patients with the diarrheal form of the disease [100].

The low FODMAPs diet also raises some doubts as to its safety; it is difficult to implement and requires a great deal of nutritional knowledge. Giving up foods high in FODMAPs carries the risk of nutrient deficiencies, such as complex carbohydrates, calcium, iron, zinc, folic acid, vitamin D, as well as compounds with antioxidant properties, such as flavonoids, carotenoids, anthocyanins and phenolic acids. Ensuring the proper intake of food products that compensate for these deficiencies is a key part of balancing a low FODMAP diet. It has also been shown that the intake of natural prebiotics, primarily fructo-oligosaccharides, galacto-oligosaccharides and fiber, is reduced in low FODMAP eaters [9,101]. This may adversely affect the intestinal microbiota, the production of short-chain fatty acids and their protective effect on colonocytes. Some papers have shown a lower concentration of *Bifidobacteria* spp., *Akkermansia muciniphila* and *Clostridium cluster XIVa* [85,86,87,90]. In contrast, other studies have not confirmed such changes, revealing no effect of a low FODMAP diet on microbiome diversity [102,103]. A study by Harvie et al. found no differences in the microbiome of IBS patients three months after the introduction of the low FODMAPs diet as compared to the standard dietary model they had followed before [102]. On the other hand, a study by McIntosh et al. demonstrated an increase in *Actinobacteria* spp. after a low FODMAP diet and a decrease in gas-producing microbes in patients on a FODMAP-rich diet [103]. These inaccuracies may be due to the variety of diets used, the choice of specific low FODMAP foods, or differences in the methodology for determining intestinal microbiota (using the sample directly or freezing it). Low FODMAP diets, due to the elimination of products that are a rich source of energy, predispose individuals to the underconsumption of energy, thereby increasing the risk of malnutrition, which often occurs in patients with IBD. This becomes particularly important in individuals with eating disorders (anorexia nervosa, orthorexia nervosa). Therefore, the proper nutrition and energy balance of a low FODMAP diet in IBD patients is essential.

There is still a lack of studies evaluating the long-term effect of introducing the low FODMAPs diet in patients with IBD. This is even more important since the low FODMAPs diet is a short-term diet regime that requires strict adherence and a subsequent return to standard nutrition. Most of the cited studies are observations lasting several weeks, conducted in small groups, based on disease symptoms. The effects of the low FODMAPs diet in IBD patients reported in some papers may result from the beneficial impact of this diet on symptoms frequently accompanying IBS [104]. Possible modifications of the low FODMAPs diet should also be discussed, especially in terms of the duration of its restrictive phase, which should be individually tailored to the patient, with the support of a doctor and a dietician.

## 5. Specific Carbohydrates Diet and Anti-Inflammatory Diet

The Specific Carbohydrates Diet (SCD) is an elimination diet that excludes complex carbohydrates and disaccharides, and requires the avoidance of wheat grains, oats, barley, corn, quinoa and rice, dairy products (except for hard cheeses and lacto-fermented products). and the substitution of sugar with honey [105]. Complex carbohydrates and disaccharides, as components that are difficult to digest, contribute to the development of dysbiosis, disruption of the intestinal barrier and development of inflammation. The intestinal barrier is a structure that is jointly formed by the microbiota, a single layer of epithelial cells, and the circulatory, lymphatic, enteric nervous and immune systems located within the lamina propria associated with the intestinal mucosa (gut-associated lymphoid tissue or GALT) [106]. Specialized connections between the cells of the epithelial monolayer provide the intestinal barrier with selective permeability to particles of a certain size and molecular charge, which consequently inhibits the development of inflammation caused by unwanted antigens from the intestinal lumen entering the vicinity of GALT. A number of dysbiotic factors, including an improperly balanced diet (too little fiber, too much saturated fat and simple sugars), antipsychotic drugs, antibiotics, proton pump inhibitors or stress, disrupt the amount, diversity, composition and function of the intestinal microbiota [28]. This dysbiosis leads to an increased permeability of the intestinal barrier. Antigens in the intestinal lumen overcome the intercellular space and activate GALT. The produced cells and effector mediators of the immune system localized in the intestine cause subclinical inflammation in situ, however, along with the blood, they reach other tissues and organs [107]. The restriction or complete exclusion of disaccharides and complex carbohydrates is therefore supposed to inhibit progressive intestinal dysbiosis, thereby mitigating the body’s immune response [105,106].

The specific carbohydrates diet is a relatively well-regarded diet among IBD patients, especially children. In a study by Cohen et al., using capsule endoscopy, intestinal mucosal improvement was observed in eight of nine children with Crohn’s disease who followed the SCD for 12 to 52 weeks. They also observed a reduction in the Harvey Bradshaw Index (HBI), from 3.3 ± 2.0 to 0.6 ± 1.3 (*p* = 0.007), and Pediatric Crohn’s Disease Activity Index (PCDAI), from 21.1 ± 5.9 to 7.8 ± 7.1 (*p* = 0.011) [76]. In a study by Obih et al. that evaluated the effect of the SCD on IBD activity in 26 children with CD and UC, a decrease in the PCDAI, from 32.8 ± 13.2 to 8.8 ± 8.5 in 20 CD patients, and a decrease in the Pediatric Ulcerative Colitis Activity Index (PCUID), from 28.3 ± 10.3 to 18.3 ± 31.7 in UC patients, were observed over six months of dietary management. CRP levels also decreased after SCD treatment in all the subjects [108]. A study by Suskind et al. showed an improvement in laboratory indices (albumin levels, hematocrit, CRP, fecal calprotectin) in all children with CD within three months of following the SCD. Patients participating in the study did not require immunosuppressive drugs. The intestinal microbiota was also assessed using DNA analysis of the microorganisms collected from stool samples. Beneficial changes in the microbiome were observed in all of them. A decrease in the concentration of *Bacteroides* spp. and *Parabacteroides* spp., and an increase in the amount of *Eubacterium* spp., *Ruminoccocus* spp. and *Subdoligranulum* spp. were found [109]. Suskind et al. also evaluated the effect of different versions of the SCD on IBD activity in a group of ten patients. They analyzed the typical SCD, its modified version, the mSCD (a diet that eliminates potatoes, rice, quinoa and oats) and the Whole Foods (WF) diet, i.e., a basic diet that excludes certain products (wheat, corn, sugar, milk and processed foods). Regardless of the elimination diet used, the reoccurrence of clinical remission was demonstrated in all the children, as well as a decrease in CRP levels [110].

A survey by Suskind et al. examined the effectiveness of the SCD based on a subjective assessment of IBD patients. The study included 417 IBD patients who declared themselves to be SCD eaters. Of the subjects, 33% reported clinical remission after two months of SCD implementation, and 42% after using the diet for six to twelve months. Among the patients reporting clinical remission, 13% observed an alleviation of disease symptoms after just two weeks of SCD, 17% after a month, 36% after up to three months, and 34% after three months after starting the diet. Among those who achieved clinical remission, 47% of patients also reported an improvement in laboratory indicators [111].

An anti-inflammatory diet pattern has been created, which is a modification of the SCD. It is based on restricting pro-inflammatory carbohydrates such as refined sugars, lactose and most grain-derived carbohydrates, and introducing plenty of anti-inflammatory products (prebiotics and probiotics). Probiotics reduce inflammation in IBD by favorably altering the microbiota, inhibiting the proliferation of pathogenic intestinal bacteria, and improving and restoring the function of the epithelial and mucosal barrier. Since probiotic strains usually do not colonize the colon, repeated and prolonged administration with long-term follow-up of the effects of such a therapy is therefore required to achieve a lasting benefit. Studies evaluating the impact of probiotics on CD activity are scarce, based on small groups of subjects, which indicates the need for further research in this area. A study conducted in a group of patients with CD showed no effect of *Lactobacillus rhamnosus GG* on sustaining remission and improving mucosal healing [112]. A study by Fedorak et al. found no statistical difference in endoscopic recurrence rates at day 90 of use between patients who received VSL#3 and those who received placebo. Lower mucosal inflammatory cytokine levels and lower recurrence rates among patients who applied VSL#3 for one year indicate that this probiotic should be further investigated for the prevention of CD recurrence [113]. Studies evaluating the effect of administered probiotic preparations in patients with UC are also inconsistent. A study by Altun et al., including a group of 40 patients with UC following a therapy with a prebiotic (fructo-oligosaccharides) and a probiotic (*Enterococcus faecium*, *Lactobacillus Plantarum*, *Streptococcus thermophilus*, *Bifidobacterium lactis*, *Lactobacillus acidophilus*, *Bifidobacterium long*), showed decreased levels of CRP, as well as clinical and endoscopic remission in all subjects [114]. Meanwhile, a randomized trial analyzing the effects of *E. coli Nissle 1917*, *S. boulardii*, *Bifidobacterium breve* and *B. bifidum* on maintaining remission in UC patients showed the same efficacy and safety of the aforementioned probiotics as the administration of mesalazine alone [115]. In the absence of conclusive study results, the American Gastroenterological Association (AGA) recommends the use of probiotics in the treatment of pouchitis. It recommends using a combination of eight strains (*L. paracasei* DSM 24733, *L. plantarum* DSM 24 730, *L. acidophilus* DSM 24735, *L. delbrueckii subsp. bulgaricus* DSM 24734, *B. longum* DSM 24736, *B. infantis* DSM 24737, *B. breve DSM 24732* and *S. thermophilus DSM 247*). The American Gastroenterological Association, however, has not put forward any guidelines for the use of probiotics in all patients with IBD, and recommends their intake only in the context of clinical trials [116]. Nevertheless, regardless of the supplementation used and its unclear effects on the microbiome, an adequate supply of probiotics and prebiotics through diet can restore the balance of bacterial flora [28,117,118,119,120,121]. The anti-inflammatory diet is high in these compounds and studies have shown its positive effects on IBD. In a study by Olendzki et al., the implementation of the diet in patients with Crohn’s disease allowed 60% of them to reduce their doses of medications used [120]. Another study showed a significant effect of the anti-inflammatory diet on the restoration of the intestinal microflora, including an increase in the levels of *Faecalibacterium prausnitzii*, *Eubacterium eligens and Roseburia hominis*, which are often found in too low amounts in patients with IBD [27].

An important problem of the SCD and the anti-inflammatory diet is the risk of nutrient deficiencies, such as iron, calcium, B vitamins and vitamin D, resulting from the complete elimination of some food groups that are rich sources of these nutrients. This is particularly important for children, in whom the aforementioned deficiencies can delay growth and development, and lead to malnutrition. Due to the highly restrictive nature of elimination diets, they should be used periodically.

## 6. Mediterranean Diet

For years, the Mediterranean diet has been recognized as a model, especially for the prevention of chronic diseases, primarily cardiovascular disease and malignancies. In 1958, the six-year Seven Countries Study (involving Japan, Greece, Yugoslavia, Italy, the Netherlands, the United States and Finland) was undertaken. It showed that the Mediterranean diet was effective in the prevention of ischemic heart disease, with a lower incidence in Greece, Italy and Japan, as compared to Finland, the United States and the Netherlands. Additionally, it was proven to result in a lower incidence of cancer, hypertension, obesity and type 2 diabetes, as well as overall mortality [121]. Nowadays, the Mediterranean diet is regarded as a dietary model that guarantees health maintenance, and recent studies also indicate its association with the prevention of allergic and neurodegenerative diseases [122,123].

The traditional Mediterranean diet is characterized by a high intake of raw vegetables and fruits, unsaturated fatty acids (mainly from olive oil and nuts), dry pulses, dairy products and fish, with a low intake of red meat and processed foods, rich in saturated fatty acids and simple sugars. Because of its high content of compounds with antioxidant properties, such as vitamins A, C, β-carotene, minerals and flavonoids, the Mediterranean diet may have anti-inflammatory effects. A study by Lewis et al. compared the effects of SCD and the Mediterranean diet on the course of the disease in patients with CD. They found that after six weeks, both the SCD and the Mediterranean diet led to clinical remission (Crohn’s Disease Activity Index, CDAI < 150) in nearly half of the patients. The remission persisted after 12 weeks of dietary modifications. In addition, there was a 35% reduction in calprotectin levels, but no significant changes in the gut microbiome of patients occurred [124]. Meanwhile, a study by Marlow et al. demonstrated the effect of the Mediterranean dietary model on the reduction in inflammatory markers and normalization of the intestinal microbiota of patients with IBD [125]. Similar data were also obtained in other studies, which observed a reduction in inflammation and the normalization of the intestinal microbiota in patients following a Mediterranean diet [126,127,128]. In addition, the Mediterranean model, apart from the influence of diet-related elements and a high intake of anti-inflammatory components, is inextricably linked to a health-promoting lifestyle and its constituents, such as regular physical activity, non-smoking and the consumption of small amounts of alcohol (primarily red wine). This affects the overall condition of the body and facilitates the maintenance of a normal body weight, which has a key impact on health status.

Apart from its anti-inflammatory properties, an important component of the Mediterranean diet in IBD patients is its high fiber content, which promotes the formation of normal intestinal microflora and the release of short-chain fatty acids which are thought to have a protective effect [48,49,50,51]. Their presence in the diet not only provides a longer feeling of satiety in the stomach, but also shortens intestinal transit time. It also stimulates fermentation processes in the large intestine, thus creating a suitable substrate for the development of beneficial bacterial microflora.

Low-fiber diets are often recommended for patients with active disease, including, for example, as part of the implementation of a low FODMAP diet [10], while patients in remission are advised to follow a standard fiber intake pattern which does not differ from the recommendations for healthy individuals [129,130,131,132]. Papers evaluating the contribution of dietary fiber based on a typical intake (without supplements) have demonstrated its effects on alleviating symptoms such as diarrhea and constipation, increasing the production of short-chain fatty acids, reducing inflammation and healing endoscopic lesions [101,133]. The large prospective cohort study, Nurses’ Health Study II, evaluated the effect of food intake during adolescence and currently on the risk of CD and UC. It was shown that in a group of women characterized by a high intake of fruits and vegetables, whole grain products, fish and poultry during adolescence, the risk of developing CD was reduced by 53%. Analyzing the impact of selected products showed that the consumption of fish at the level of a minimum of 30 g per day in the past reduced the risk of CD by 57%. In contrast, the current consumption of the analyzed food groups did not significantly differentiate the risk of CD in the subjects. However, the correlations found were not confirmed for UC [134]. Similar findings were also obtained in smaller studies. A study by Brotherton et al. evaluated the effect of fiber intake on disease activity and quality of life in people with IBD. In a 28-day follow-up, patients were divided into a study group, which was advised to eat oatmeal once a day, and a control group, which received general dietary recommendations without specific indications. It was shown that in the programmed study group, fiber intake did not exacerbate disease symptoms, and patients experienced less frequent exacerbations of CD, as measured by HDI, than those in the control group, with lower dietary fiber intake. These patients also showed an improved quality of life, as measured by the Inflammatory Bowel Disease Questionnaire (IBDQ) (Table 1) [135].

## 7. IOIBD Guidelines

The International Organization for the Study of Inflammatory Bowel Disease (IOIBD) is the only international organization dedicated to the study of inflammatory bowel disease. Its mission is to promote the health of people with IBD worldwide by setting the directions for patient care, education and ongoing research. The IOIBD aims to develop clear definitions of IBD symptoms, as well as the effects of treatment in practice and clinical trials. The long-term goal of the IOIBD is to develop causal treatments for these diseases.

In the absence of clear indications for following a specific dietary model, the IOIBD has formulated dietary recommendations relating to all food groups [5]. The current recommendations, as defined by the IOIBD, were created based on the results of research in recent years. They reached a consensus for all food groups, except pasteurized dairy products.

The IOIBD recommends a moderate to high intake of vegetables and fruits. The exception concerns CD patients with existing intestinal strictures, who are recommended to limit their intake of the insoluble fraction of dietary fiber, thus reducing their intake of certain vegetables (cauliflower, Chinese cabbage, spinach, tomatoes, dry pulses) and fruits (raspberries, gooseberries, kiwi, avocados). Cereal products are also an important dietary source of fiber. The IOIBD advises the consumption of complex carbohydrates and simple sugars consistent with recommendations for healthy individuals. The exception includes CD and UC patients with functional bowel disorders, without active inflammation. In these patients, a reduced FODMAP intake is suggested. There are also no indications for restricting wheat and gluten protein intake in patients with IBD. There are studies that have found the alleviation of disease symptoms in patients who gave up gluten. However, this effect was attributed to a simultaneous lower intake of FODMAPs through the diet.

The IOIBD has not introduced restrictions on the consumption of red meat, poultry or eggs in people with CD. However, patients with UC should reduce their intake of red meat and its products due to their high content of saturated fatty acids, especially myristic acid. Studies have shown a correlation between high red meat consumption and the incidence of relapse in patients with UC, while this relationship has not been confirmed in patients with CD [136,137,138,139,140,141]. In the IOIBD guidelines, special attention was given to the quality of dietary fats. The need to eliminate trans fatty acids and saturated fatty acids in all IBD patients was emphasized. In addition, a recommendation to reduce the intake of myristic acid (palm fat, coconut fat, beef, dairy products) and increase the intake of *n-3* fatty acids in patients with UC was introduced.

While the IOIBD has unequivocally advocated against the consumption of milk and unpasteurized products, no clear recommendations have been established in regards to the consumption of pasteurized dairy products in IBD patients. Studies have shown a higher prevalence of lactose intolerance in people with CD and UC than in healthy subjects [57,141], which supports the elimination of all milk-derived products. However, due to the frequent presence of additives such as emulsifiers, carrageenans and thickening agents in dairy products, it is difficult to establish clear criteria.

The IOIBD guidelines unequivocally recommend eliminating foods rich in additives such as maltodextrin, emulsifiers, thickeners, nanoparticles and sulfur compounds in all IBD patients. In practice, this means giving up processed and so-called convenience foods.

## 8. Conclusions

Certainly, the appropriate dietary management of IBD is an important therapeutic component which allows for the alleviation of symptoms, improvement of quality of life and maintenance of a sustained remission of the disease. According to both ECCO and ESPEN recommendations, there is currently no single diet that can be recommended for patients with IBD [69,70]. However, the dietary interventions discussed are fairly well-researched and can be used to treat CD and UC, depending on the presence of food intolerances, patients’ own diet history and preferences (Table 2).

The specific carbohydrate diet is probably the most thoroughly discussed dietary management in patients with IBD. Apart from the elimination of some complex carbohydrates, it also implies limiting the intake of highly processed foods. Although most of the available studies on the validity of implementing the SCD in these patients are based on small trials, the results suggest its high efficacy in alleviating disease symptoms, reducing inflammation and improving the biodiversity of the intestinal microbiota. However, considering the lack of data on its effect on endoscopic indices, it is advisable to monitor the mucosal condition in patients, even those in clinical remission [117].

In patients with IBD and accompanying functional bowel disorders, the most recommended diet is low FODMAP. It has been shown in studies to be effective in relieving symptoms such as abdominal pain, bloating and diarrhea, however, it does not lower inflammatory markers. Therefore, it can be recommended to patients with inactive IBD and intestinal dysfunction, but without coexisting inflammation. The shortcoming of this diet is the high risk of adverse changes in the intestinal microbiome, due to the restriction of complex carbohydrate intake. Therefore, the low FODMAPs diet should not be a long-term diet. In addition, it is recommended that the elimination of FODMAPs be introduced gradually, in order to determine the individual sensitivity of the patient to their particular groups. This can prevent the unnecessary elimination of ingredients well-tolerated by the patient.

In patients without known food intolerances and allergies, the Mediterranean diet appears to be the safest eating model, due to its very high nutritional value and high anti-inflammatory potential. Despite the few results of studies investigating the effectiveness of the Mediterranean diet among patients with IBD, it is a very well-studied dietary model. The Mediterranean diet is varied, based on products with high nutritional value, thus ensuring the implementation of dietary standards for all nutrients. The undoubted advantage of the Mediterranean diet is the simplicity of its application and reliance on commonly available products.

Despite the variety of diets studied in IBD patients, there are no clear recommendations for applying one specific diet in all patients. None of the diets discussed should be guided in a universal manner due to the differences in nutritional status, tolerance of different food groups and dietary pattern of patients. It is also important to confirm food intolerances before introducing any elimination diet. The most commonly indicated food allergies involve gluten and lactose, however, these allergies should necessarily be documented.

Although there are no clear guidelines for selecting a specific dietary pattern or food product, the common element of all recommendations is to avoid processed foods, products rich in food additives and containing high amounts of saturated and trans fatty acids. Therefore, these recommendations should be given to all patients with IBD.

## Figures and Tables

**Table 1 nutrients-14-04261-t001:** Dietary interventions in IBD patients.

Diet	Study	Population	Number of Subjects	Duration	Results
Low FODMAP	Bodini (2019) Randomized trial [90]	Adults with CD and UC in remission or with mild-to-moderate disease	*n* = 55	6 weeks	Reduced gastrointestinal symptoms, reduced fecal calprotectin, improved quality of life
	Cox (2020) Placebo-controlled trial [136]	Adults with CD and UC in remission	*n* = 52 (27 low FODMAP, 25 placebo diet)	4 weeks	Improved quality of life, reduced *Bifidobacterium adolescentis*, *Bifidobacterium longum* and *Faecalibacterium prausnitzii*, reduced SCFA
	Pedersen (2017) Randomized controlled open-label trial [137]	Adults with CD and UC in remission or with mild-to-moderate disease and coexisting IBS-like symptoms	*n* = 89	6 weeks	Improved quality of life, reduced IBS-like symptoms
	Halmos (2016) Randomized, controlled cross-over trial [83]	Adults with CD	*n* = 9	3 weeks	Reduced gastrointestinal symptoms
GFD	Herfarth (2014) Cross-sectional study [75]	Adults with CD and UC	*n* = 1647	-	Improved gastrointestinal symptoms, fewer or less severe IBD flares
	Schreiner (2019) Cohort study [138]	Adults with CD and UC	*n* = 1254	-	Reduced amount and biodiversity of microbiota, lower psychological well-being
	Lopes (2022) Prospective cohort study [78]	Adults with CD and UC	*n* = 784	-	No association with risk of CD or UC
SCD	Suskind (2014) Retrospective study [103]	Children with CD	*n* = 7	5–30 months	Reduced symptoms, reduced fecal calprotectin, reduced inflammation
	Suskind (2018) Prospective study [106]	Children with CD	*n* = 12	12 weeks	Reduced symptoms, reduced inflammation, improved microbial composition
	Suskind (2020) Randomized diet-controlled trial [107]	Children with CD	*n* = 18	12 weeks	Reduced symptoms, reduced inflammation, improved microbiome composition
	Cohen (2014) Prospective study [73]	Children with CD	*n* = 16	52 weeks	Clinical and mucosal improvements
MD	Lewis (2021) Randomized trial [120]	Adult active CD	*n* = 191 (99 SCD, 92 MD)	12 weeks	Reduced fecal calprotectin, reduced *Eubacterium* spp. and *Faecalibacterium prausnitzii*, increased abundance of *Bacteroides vulgatus* and *Enterobacteriaceae* spp.
Anti-Inflammatory diet	Olendzki (2014) Case series report [116]	Inactive or mild adult with CD and UC	*n* = 40	4 weeks	Reduced symptoms, reduced inflammation

**Table 2 nutrients-14-04261-t002:** Evidence-based diet recommendations for the maintenance of remission in IBD.

Diet	Characteristics	Recommendation
Lactose-free diet [44,45,46,47,48,49,50,51,52,53,54,55,56,57,58,59,60,61,62,63,64,65,66,67,68,69,70,71,72,73]	Exclusion of diet products with lactose: milk, milk products, butter, dried milk, cheese, cream, milk-based nutritional supplements, sugar substitutes with lactose added, medications and vitamin/mineral supplements with lactose added	No recommendation *
Gluten-free diet [74,75,76,77,78,79,80,81,82,83]	Exclusion of diet foods containing gluten: wheat, rye, barley, triticale, and oats contaminated during production with wheat, barley or rye	No recommendation *
SCD [27,76,105,106,107,108,109,110,111,112,113,114,115,116]	Exclusion of dietary complex carbohydrates and disaccharides, wheat grains, oats, barley, corn, quinoa, rice and dairy products (except for hard cheeses and lacto-fermated products)	Optional **
Low FODMAP [61,75,84,85,86,87,88,89,90,91,92,93,94,95,96,97,98,99]	Exclusion of dietary high FODMAP products: dairy-based milk, yogurt, ice cream, wheat-based products such as cereal, bread and crackers, beans and lentils, vegetables, such as artichokes, asparagus, onions and garlic, and fruits, such as apples, cherries, pears and peaches.	Optional **
MD [48,49,50,51,122,123,124,125,126,127,128,129,130,131,132,133,134,135]	High intake of raw vegetables and fruits, unsaturated fatty acids (mainly from olive oil and nuts), dry pulses, dairy products and fish; Low intake of red meat and processed foods, rich in saturated fatty acids, and simple sugars.	Optional **
Anti-inflammatory diet [112,113,114,115,116,117,118,119,120,121]	High intake of foods high in antioxidants: apples, artichokes, avocados, beans (such as red beans, pinto beans and black beans), berries (such as blueberries, raspberries and blackberries), broccoli, cherries, dark chocolate (at least 70% cocoa), dark green leafy vegetables (such as kale, spinach and collard greens), nuts (such as walnuts, almonds, pecans and hazelnuts), sweet potatoes and whole grains; high intake of foods high in omega-3 fatty acids: flaxseed, oily fish (salmon, herring, mackerel, sardines and anchovies), omega-3-fortified foods (including eggs and milk) and walnuts; Low intake of foods high in omega-6 fatty acids: high-fat dairy products (milk, cheese, butter and ice cream), margarine, meats and peanuts	Optional **

* No recommendation: There is a lack of or poor evidence. There is an unclear balance between benefit and harm. ** Optional: The quality of evidence is suspect, further studies are needed. It may be of limited advantage.

## Data Availability

Not applicable.

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
