# Peer review of "Dietary Interventions in Inflammatory Bowel Disease"

_nutrients, 2022, doi:10.3390/nu14204261_

Round 1

Reviewer 1 Report

The authors clearly demonstrated the association between various dietary interventions and the disease conditions of IBD.  In particular, it is interesting to note that the authors focused on dietary style rather than dietary component.  In general, I believe that this is well-written paper including several interesting data. As I have a couple of concerns, I hope the authors check the following issues.

1. The authors should demonstrate the detailed characteristics in each dietary style.

2. In conclusion, the authors summarized their views. I recommend a tabular summary of these opinions for the reader's better understanding. 

Author Response

Dear Reviewer,

Thank you for giving us the opportunity to submit a revised draft of the manuscript. We
appreciate the time and effort that you dedicated to providing feedback on our
manuscript and are grateful for the insightful comments on and valuable improvements to our paper. We have incorporated most of the suggestions made by the reviewers. Those changes are highlighted within the manuscript.

We demonstrated the detailed characteristic of each dietary style in table 2.

We summarized our opinions in table 2 to make it more clear for readers.

Reviewer 2 Report

The authors of this study on dietary interventions in inflammatory bowel disease (IBD) sought to assess the dietary models used in research studies and their potential influence on IBD activity.

The study that has been presented offers important details on how diets affect IBD activity. But I do have a few remarks and advice.

The introduction should be condensed because it is too wordy and unclear. I believe that the Authors did not need to consider the diet's contribution to the etiology of IBD. In actuality, they must assess how those influences affect disease activity. As a result, authors should concentrate on how nutrition affects the severity and activity of IBD in the introduction and subsequent chapters (Lactose-free diet, Gluten-free diet, Low-FODMAP diet, Specific carbohydrates diet and anti-inflammatory diet, and Mediterranean diet). However, the lactose-free diet section contains some information on both the causes and progression of IBD. The information in the text concerning diet-causing IBD should be removed.

Author Response

Dear Reviewer,

Thank you for giving us the opportunity to submit a revised draft of the manuscript. We
appreciate the time and effort that you dedicated to providing feedback on our
manuscript and are grateful for the insightful comments on and valuable improvements to our paper. We have incorporated most of the suggestions made by the reviewers. Those changes are highlighted within the manuscript.

We made the introduction shorter by withdrawing the part concerning the impact of diet on intestinal microbiota.

We summarized our opinions about diet recommendations in table 2 for the readers’ better understanding.

We did not removed the information about diet causing IBD in the part concerning lactose-free diet, because in our opinion it is important part of the discussion.